# Flexible and efficient simulation-based inference for models of decision-making

Jan Boelts[1,2]*, Jan-Matthis Lueckmann[1], Richard Gao[1], Jakob H Macke[1,3]

[1]Machine Learning in Science, Excellence Cluster Machine Learning, University of Tübingen, Tübingen, Germany; [2]Technical University of Munich, Munich, Germany; [3]Max Planck Institute for Intelligent Systems Tübingen, Tübingen, Germany

**Abstract** Inferring parameters of computational models that capture experimental data is a central task in cognitive neuroscience. Bayesian statistical inference methods usually require the ability to evaluate the likelihood of the model—however, for many models of interest in cognitive neuroscience, the associated likelihoods cannot be computed efficiently. Simulation-based inference (SBI) offers a solution to this problem by only requiring access to simulations produced by the model. Previously, Fengler et al. introduced likelihood approximation networks (LANs, Fengler et al., 2021) which make it possible to apply SBI to models of decision-making but require billions of simulations for training. Here, we provide a new SBI method that is substantially more simulation efficient. Our approach, mixed neural likelihood estimation (MNLE), trains neural density estimators on model simulations to emulate the simulator and is designed to capture both the continuous (e.g., reaction times) and discrete (choices) data of decision-making models. The likelihoods of the emulator can then be used to perform Bayesian parameter inference on experimental data using standard approximate inference methods like Markov Chain Monte Carlo sampling. We demonstrate MNLE on two variants of the drift-diffusion model and show that it is substantially more efficient than LANs: MNLE achieves similar likelihood accuracy with six orders of magnitude fewer training simulations and is significantly more accurate than LANs when both are trained with the same budget. Our approach enables researchers to perform SBI on custom-tailored models of decision-making, leading to fast iteration of model design for scientific discovery.

**\*For correspondence:**
jan.boelts@uni-tuebingen.de

**Competing interest:** The authors declare that no competing interests exist.

## Editor's evaluation

This paper provides a new approach, Mixed Neural Likelihood Estimator (MNLE) to build likelihood emulators for simulation-based models where the likelihood is unavailable. The authors show that the MNLE approach is equally accurate but orders of magnitude more efficient than a recent proposal, likelihood approximation networks (LAN), on two variants of the drift-diffusion model (a widely used model in cognitive neuroscience). This work provides a practical approach for fitting more complex models of behavior and neural activity for which likelihoods are unavailable.

## Introduction

Computational modeling is an essential part of the scientific process in cognitive neuroscience: Models are developed from prior knowledge and hypotheses, and compared to experimentally observed phenomena (*Churchland and Sejnowski, 1988*; *McClelland, 2009*). Computational models usually have free parameters which need to be tuned to find those models that capture experimental data. This is often approached by searching for single best-fitting parameters using grid search or optimization methods. While this point-wise approach has been used successfully (*Lee et al., 2016*; *Patil et al., 2016*) it can be scientifically more informative to perform Bayesian inference over the

model parameters: Bayesian inference takes into account prior knowledge, reveals *all* the parameters consistent with observed data, and thus can be used for quantifying uncertainty, hypothesis testing, and model selection (*Lee, 2008*; *Shiffrin et al., 2008*; *Lee and Wagenmakers, 2014*; *Schad et al., 2021*). Yet, as the complexity of models used in cognitive neuroscience increases, Bayesian inference becomes challenging for two reasons. First, for many commonly used models, computational evaluation of likelihoods is challenging because often no analytical form is available. Numerical approximations of the likelihood are typically computationally expensive, rendering standard approximate inference methods like Markov Chain Monte Carlo (MCMC) inapplicable. Second, models and experimental paradigms in cognitive neuroscience often induce scenarios in which inference is repeated for varying numbers of experimental trials and changing hierarchical dependencies between model parameters (*Lee, 2011*). As such, fitting computational models with arbitrary hierarchical structures to experimental data often still requires idiosyncratic and complex inference algorithms.

Approximate Bayesian computation (ABC, *Sisson, 2018*) offers a solution to the first challenge by enabling Bayesian inference based on comparing simulated with experimental data, without the need to evaluate an explicit likelihood function. Accordingly, various ABC methods have been applied to and developed for models in cognitive neuroscience and related fields (*Turner and Van Zandt, 2012*; *Turner and Van Zandt, 2018*; *Palestro et al., 2009*; *Kangasrääsiö et al., 2019*). However, ABC methods are limited regarding the second challenge because they become inefficient as the number of model parameters increases (*Lueckmann et al., 2021*) and require generating new simulations whenever the observed data or parameter dependencies change.

More recent approaches from the field simulation-based inference (SBI, *Cranmer et al., 2020*) have the potential to overcome these limitations by using machine learning algorithms such as neural networks. Recently, *Fengler et al., 2021* presented an SBI algorithm for a specific problem in cognitive neuroscience—inference for drift-diffusion models (DDMs). They introduced a new approach, called likelihood approximation networks (LANs), which uses neural networks to predict log-likelihoods from data and parameters. The predicted likelihoods can subsequently be used to generate posterior samples using MCMC methods. LANs are trained in a three-step procedure. First, a set of $N$ parameters is generated and for each of the $N$ parameters the model is simulated $M$ times. Second, for each of the $N$ parameters, empirical likelihood targets are estimated from the $M$ model simulations using kernel density estimation (KDE) or empirical histograms. Third, a training dataset consisting of parameters, data points, and empirical likelihood targets is constructed by augmenting the initial set of $N$ parameters by a factor of 1000: for each parameter, 1000 data points and empirical likelihood targets are generated from the learned KDE. Finally, supervised learning is used to train a neural network to predict log-likelihoods, by minimizing a loss function (the Huber loss) between the network-predicted log-likelihoods and the (log of) the empirically estimated likelihoods. Overall, LANs require a large number of model simulations such that the histogram probability of each possible observed data and for each possible combination of input parameters, can be accurately estimated—$N \cdot M$ model simulations, for example, 1.5 (150 billion) for the examples used in *Fengler et al., 2021*. The extremely high number of model simulations will make it infeasible for most users to run this training themselves, so that there would need to be a repository from which users can download pretrained LANs. This restricts the application of LANs to a small set of canonical models like DDMs, and prohibits customization and iteration of models by users. In addition, the high simulation requirement limits this approach to models whose parameters and observations are sufficiently low dimensional for histograms to be sampled densely.

To overcome these limitations, we propose an alternative approach called mixed neural likelihood estimation (MNLE). MNLE builds on recent advances in probabilistic machine learning, and in particular on the framework of neural likelihood estimation (*Papamakarios et al., 2019b*; *Lueckmann et al., 2019*) but is designed to specifically capture the mixed data types arising in models of decision-making, for example, discrete choices and continuous reaction times. Neural likelihood estimation has its origin in classical synthetic likelihood (SL) approaches (*Wood, 2010*; *Price et al., 2018*). Classical SL approaches assume the likelihood of the simulation-based model to be Gaussian (so that its moments can be estimated from model simulations) and then use MCMC methods for inference. This approach and various extensions of it have been widely used (*Price et al., 2018*; *Ong et al., 2009*; *An et al., 2019*; *Priddle et al., 2022*)—but inherently they need multiple model simulations for each parameter in the MCMC chain to estimate the associated likelihood.

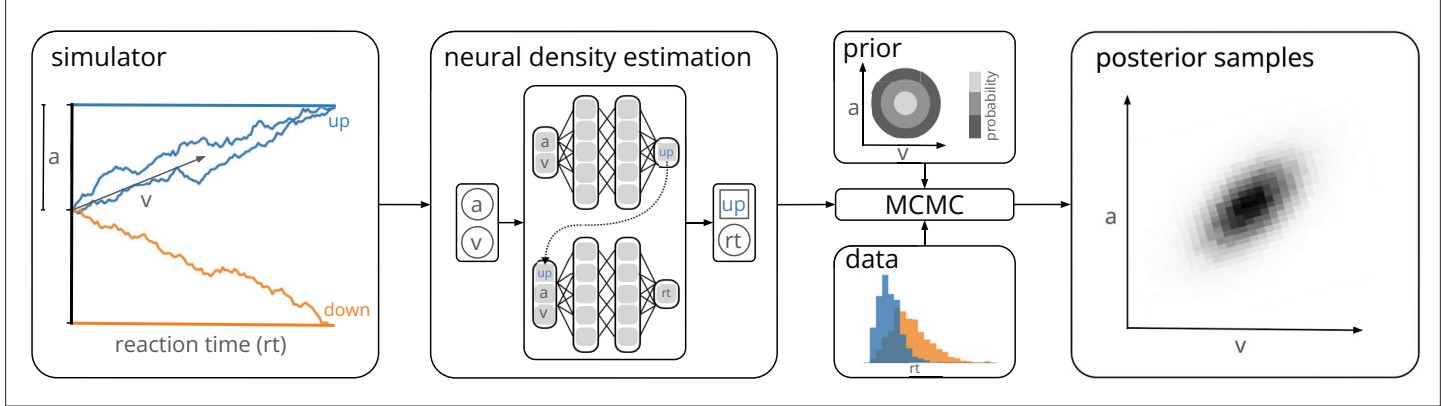

**Figure 1.** Training a neural density estimator on simulated data to perform parameter inference. Our goal is to perform Bayesian inference on models of decision-making for which likelihoods cannot be evaluated (here a drift-diffusion model for illustration, left). We train a neural density estimation network on synthetic data generated by the model, to provide access to (estimated) likelihoods. Our neural density estimators are designed to account for the mixed data types of decision-making models (e.g., discrete valued choices and continuous valued reaction times, middle). The estimated likelihoods can then be used for inference with standard Markov Chain Monte Carlo (MCMC) methods, that is, to obtain samples from the posterior over the parameters of the simulator given experimental data (right). Once trained, our method can be applied to flexible inference scenarios like varying number of trials or hierarchical inference without having to retrain the density estimator.

*Neural* likelihood approaches instead perform *conditional* density estimation, that is, they train a neural network to predict the parameters of the approximate likelihood conditioned on the model parameters (e.g., *Papamakarios et al., 2019b*; *Lueckmann et al., 2019*). By using a conditional density estimator, it is possible to exploit continuity across different model parameters, rather than having to learn a separate density for each individual parameter as in classical SL. Recent advances in conditional density estimation (such as normalizing flows, *Papamakarios et al., 2019a*) further allow lifting the parametric assumptions of classical SL methods and learning flexible conditional density estimators which are able to model a wide range of densities, as well as highly nonlinear dependencies on the conditioning variable. In addition, neural likelihood estimators yield estimates of the probability density which are guaranteed to be non-negative and normalized, and which can be both sampled and evaluated, acting as a probabilistic emulator of the simulator (*Lueckmann et al., 2019*).

Our approach, MNLE, uses neural likelihood estimation to learn an emulator of the simulator. The training phase is a simple two-step procedure: first, a training dataset of $N$ parameters $\boldsymbol{\theta}$ is sampled from a proposal distribution and corresponding model simulations $\mathbf{x}$ are generated. Second, the $N$ parameter–data pairs $(\boldsymbol{\theta}, \mathbf{x})$ are directly used to train a conditional neural likelihood estimator to estimate $p(\mathbf{x}|\boldsymbol{\theta})$. Like for LANs, the proposal distribution for the training data can be *any* distribution over $\boldsymbol{\theta}$, and should be chosen to cover all parameter values one expects to encounter in empirical data. Thus, the prior distribution used for Bayesian inference constitutes a useful choice, but in principle any distribution that contains the support of the prior can be used. To account for mixed data types, we learn the likelihood estimator as a mixed model composed of one neural density estimator for categorical data and one for continuous data, conditioned on the categorical data. This separation allows us to choose the appropriate neural density estimator for each data type, for example, a Bernoulli model for the categorical data and a normalizing flow (*Papamakarios et al., 2019a*) for the continuous data. The resulting joint density estimator gives access to the likelihood, which enables inference via MCMC methods. See *Figure 1* for an illustration of our approach, and Methods and materials for details.

Both LANs and MNLEs allow for flexible inference scenarios common in cognitive neuroscience, for example, varying number of trials with same underlying experimental conditions or hierarchical inference, and need to be trained only once. However, there is a key difference between the two approaches. LANs use feed-forward neural networks to perform regression from model parameters to empirical likelihood targets obtained from KDE. MNLE instead learns the likelihood directly by performing conditional density estimation on the simulated data without requiring likelihood targets. This makes MNLE by design more simulation efficient than LANs—we demonstrate empirically that it can learn likelihood estimators which are as good or better than those reported in the LAN paper,

but using a factor of 1,000,000 fewer simulations (*Fengler et al., 2021*). When using the same simulation budget for both approaches, MNLE substantially outperforms LAN across several performance metrics. Moreover, MNLE results in a density estimator that is guaranteed to correspond to a valid probability distribution and can also act as an emulator that can generate synthetic data without running the simulator. The simulation efficiency of MNLEs allows users to explore and iterate on their own models without generating a massive training dataset, rather than restricting their investigations to canonical models for which pretrained networks have been provided by a central resource. To facilitate this process, we implemented our method as an extension to an open-source toolbox for SBI methods (*Tejero-Cantero et al., 2020*), and provide detailed documentation and tutorials.

## Results

### Evaluating the performance of MNLE on the DDM

We first demonstrate the efficiency and performance of MLNEs on a classical model of decision-making, the DDM (*Ratcliff and McKoon, 2008*). The DDM is an influential phenomenological model of a two-alternative perceptual decision-making task. It simulates the evolution of an internal decision variable that integrates sensory evidence until one of two decision boundaries is reached and a choice is made (*Figure 1*, left). The decision variable is modeled with a stochastic differential equation which, in the 'simple' DDM version (as used in *Fengler et al., 2021*), has four parameters: the drift rate $v$, boundary separation $a$, the starting point $w$ of the decision variable, and the non-decision time $\tau$. Given these four parameters $\boldsymbol{\theta} = (v, a, w, \tau)$, a single simulation of the DDM returns data $\mathbf{x}$ containing a choice $c \in \{0, 1\}$ and the corresponding reaction time in seconds $rt \in (\tau, \infty)$.

### MNLE learns accurate likelihoods with a fraction of the simulation budget

The simple version of the DDM is the ideal candidate for comparing the performance of different inference methods because the likelihood of an observation given the parameters, $L(\mathbf{x}|\boldsymbol{\theta})$, *can* be calculated analytically (*Navarro and Fuss, 2009*, in contrast to more complicated versions of the DDM, e.g., *Ratcliff and Rouder, 1998*; *Usher and McClelland, 2001*; *Reynolds and Rhodes, 2009*). This enabled us to evaluate MNLE's performance with respect to the analytical likelihoods and the corresponding inferred posteriors of the DDM, and to compare against that of LANs on a range of simulation budgets. For MNLE, we used a budget of $10^5$ simulations (henceforth referred to as MNLE[5]), for LANs we used budgets of $10^5$ and $10^8$ simulations (LAN[5] and LAN[8], respectively, trained by us) and the pretrained version based on $10^{11}$ simulations (LAN[11]) provided by *Fengler et al., 2021*.

First, we evaluated the quality of likelihood approximations of MNLE[5], and compared it to that of LAN[5,8,11]. Both MNLEs and LANs were in principle able to accurately approximate the likelihoods for both decisions and a wide range of reaction times (see *Figure 2a* for an example, and Details of the numerical comparison). However, LANs require a much larger simulation budget than MNLE to achieve accurate likelihood approximations, that is, LANs trained with $10^5$ or $10^8$ simulations show visible deviations, both in the linear and in log-domain (*Figure 2a*, lines for LAN[5] and LAN[8]).

To quantify the quality of likelihood approximation, we calculated the Huber loss and the mean-squared error (MSE) between the true and approximated likelihoods (*Figure 2b, c*), as well as between the *log*-likelihoods (*Figure 2d, e*). The metrics were calculated as averages over (log-)likelihoods of a fixed observation given 1000 parameters sampled from the prior, repeated for 100 observations simulated from the DDM. For metrics calculated on the untransformed likelihoods (*Figure 2b, c*), we found that MNLE[5] was more accurate than LAN[5,8,11] on all simulation budgets, showing smaller Huber loss than LAN[5,8,11] in 99, 81, and 66 out of 100 comparisons, and smaller MSE than LAN[5,8,11] on 98, 81, and 66 out of 100 comparisons, respectively. Similarly, for the MSE calculated on the log-likelihoods (*Figure 2e*), MNLE[5] achieved smaller MSE than LAN[5,8,11] on 100, 100, and 75 out of 100 comparisons, respectively. For the Huber loss calculated on the log-likelihoods (*Figure 2d*), we found that MNLE[5] was more accurate than LAN[5] and LAN[8], but slightly less accurate than LAN[11], showing smaller Huber loss than LAN[5,8] in all 100 comparisons, and larger Huber loss than LAN[11] in 62 out of 100 comparisons. All the above pairwise comparisons were significant under the binomial test (p < 0.01), but note that these are simulated data and therefore the p value can be arbitrarily inflated by increasing the number of comparisons. We also

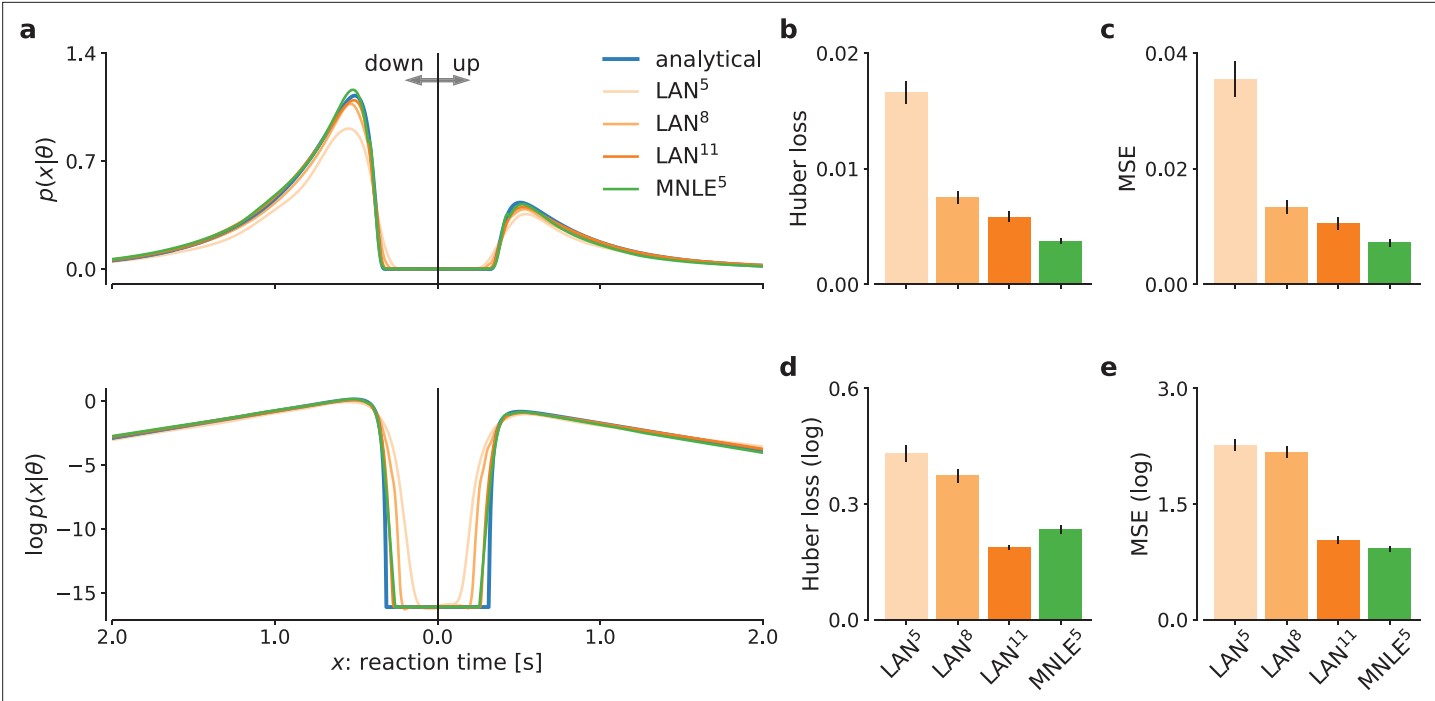

**Figure 2.** Mixed neural likelihood estimation (MNLE) estimates accurate likelihoods for the drift-diffusion model (DDM). The classical DDM simulates reaction times and choices of a two-alternative decision task and has an analytical likelihood which can be used for comparing the likelihood approximations of MNLE and likelihood approximation network (LAN). We compared MNLE trained with a budget of $10^5$ simulations (green, MNLE[5]) to LAN trained with budgets of $10^5$, $10^8$, and $10^{11}$ simulations (shades of orange, LAN[5,8,11], respectively). (**a**) Example likelihood for a fixed parameter $\boldsymbol{\theta}$ over a range of reaction times (reaction times for down- and up-choices shown toward the left and right, respectively). Shown on a linear scale (top panel) and a logarithmic scale (bottom panel). (**b**) Huber loss between analytical and estimated likelihoods calculated for a fixed simulated data point over 1000 test parameters sampled from the prior, averaged over 100 data points (lower is better). Bar plot error bars show standard error of the mean. (**c**) Same as in (**b**), but using mean-squared error (MSE) over likelihoods (lower is better). (**d**) Huber loss on the log-likelihoods (LAN's training loss). (**e**) MSE on the log-likelihoods.

The online version of this article includes the following figure supplement(s) for figure 2:

**Figure supplement 1.** Comparison of simulated drift-diffusion model (DDM) data and synthetic data sampled from the mixed neural likelihood estimation (MNLE) emulator.

note that the Huber loss on the log-likelihoods is the loss which is directly optimized by LANs, and thus this comparison should in theory favor LANs over alternative approaches. Furthermore, the MNLE[5] results shown here represent averages over 10 random neural network initializations (five of which achieved smaller Huber loss than LAN[11]), whereas the LAN[11] results are based on a single pretrained network. Finally, we also investigated MNLE's property to act as an emulator of the simulator and found the synthetic reaction times and choices generated from the MNLE emulator to match corresponding data simulated from the DDM accurately (see *Figure 2—figure supplement 1* and Appendix 1).

When using the learned likelihood estimators for inference with MCMC methods, their evaluation speed can also be important because MCMC often requires thousands of likelihood evaluations. We found that evaluating MNLE for a batch of 100 trials and 10 model parameters (as used during MCMC) took 4.14± (mean over 100 repetitions ± standard error of the mean), compared to 1.02± for LANs, that is, MNLE incurred a larger computational foot-print at evaluation time. Note that these timings are based on an improved implementation of LANs compared to the one originally presented in *Fengler et al., 2021*, and evaluation times can depend on the implementation, compute infrastructure and parameter settings (see Details of the numerical comparison and Discussion). In summary, we found that MNLE trained with $10^5$ simulations performed substantially better than LANs trained with $10^5$ or $10^8$ simulations, and similarly well or better than LANs trained with $10^{11}$ simulations, on all likelihood approximation accuracy metrics.

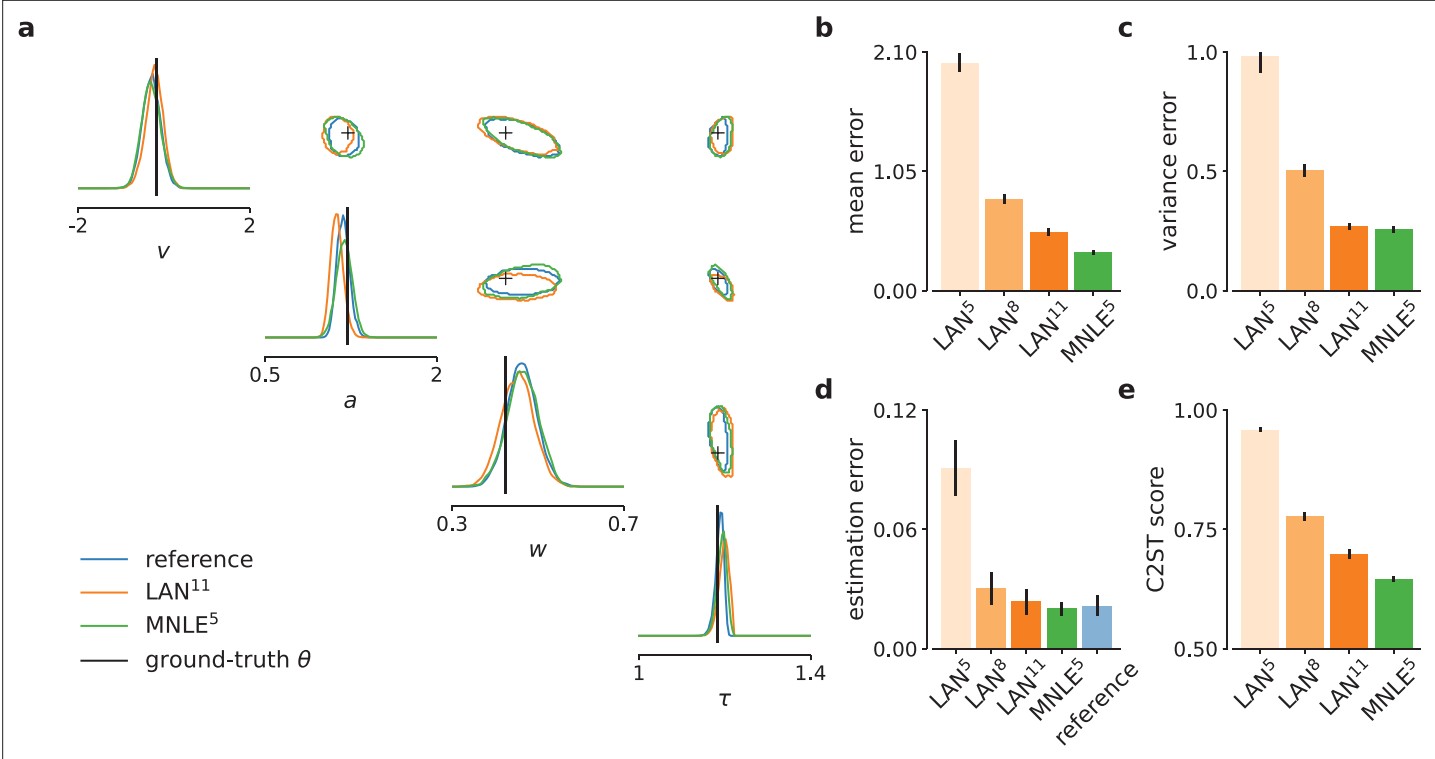

**Figure 3.** Mixed neural likelihood estimation (MNLE) infers accurate posteriors for the drift-diffusion model. Posteriors were obtained given 100-trial independent and identically distributed (i.i.d.) observations with Markov Chain Monte Carlo (MCMC) using analytical (i.e., reference) likelihoods, or those approximated using LAN[{5,8,11}] trained with simulation budgets $10^{\{5,8,11\}}$, respectively, and MNLE[5] trained with a budget of $10^5$ simulations. (**a**) Posteriors given an example observation generated from the prior and the simulator, shown as 95% contour lines in a corner plot, that is, one-dimensional marginal (diagonal) and all pairwise two-dimensional marginals (upper triangle). (**b**) Difference in posterior sample mean of approximate (LAN[{5,8,11}], MNLE[5]) and reference posteriors (normalized by reference posterior standard deviation, lower is better). (**c**) Same as in (**b**) but for posterior sample variance (normalized by reference posterior variance, lower is better). (**d**) Parameter estimation error measured as mean-squared error (MSE) between posterior sample mean and the true underlying parameters (smallest possible error is given by reference posterior performance shown in blue). (**e**) Classification 2-sample test (C2ST) score between approximate (LAN[{5,8,11}], MNLE[5]) and reference posterior samples (0.5 is best). All bar plots show metrics calculated from 100 repetitions with different observations; error bars show standard error of the mean.

The online version of this article includes the following figure supplement(s) for figure 3:

**Figure supplement 1.** Drift-diffusion model (DDM) inference accuracy metrics for individual model parameters.

**Figure supplement 2.** Drift-diffusion model (DDM) example posteriors and parameter recovery for likelihood approximation networks (LANs) trained with smaller simulation budgets.

**Figure supplement 3.** Drift-diffusion model (DDM) inference accuracy metrics for different numbers of observed trials.

## MNLE enables accurate flexible posterior inference with MCMC

In the previous section, we showed that MNLE requires substantially fewer training simulations than LANs to approximate the likelihood accurately. To investigate whether these likelihood estimates were accurate enough to support accurate parameter inference, we evaluated the quality of the resulting posteriors, using a framework for benchmarking SBI algorithms (*Lueckmann et al., 2021*). We used the analytical likelihoods of the simple DDM to obtain reference posteriors for 100 different observations, via MCMC sampling. Each observation consisted of 100 independent and identically distributed (i.i.d.) trials simulated with parameters sampled from the prior (see *Figure 3a* for an example, details in Materials and methods). We then performed inference using MCMC based on the approximate likelihoods obtained with MNLE ($10^5$ budget, MNLE[5]) and the ones obtained with LAN for each of the three simulation budgets (LAN {5,8,11}, respectively).

Overall, we found that the likelihood approximation performances presented above were reflected in the inference performances: MNLE[5] performed substantially better than LAN[5] and LAN[8], and equally well or better than LAN[11] (*Figure 3b–d*). In particular, MNLE[5] approximated the posterior

mean more accurately than LAN[5,8,11] (*Figure 3b*), being more accurate than LAN[5,8,11] in 100, 90, and 67 out of 100 comparisons, respectively. In terms of posterior variance, MNLE[5] performed better than LAN[5,8] and on par with LAN[11] (*Figure 3c*), being more accurate than LAN[5,8,11] in 100, 93 (p <<0.01, binomial test), and 58 ($p = 0.13$) out of 100 pairwise comparisons, respectively.

Additionally, we measured the parameter estimation accuracy as the MSE between the posterior mean and the ground-truth parameters underlying the observed data. We found that MNLE[5] estimation error was indistinguishable from that of the reference posterior, and that LAN performance was similar only for the substantially larger simulation budget of LAN[11] (*Figure 3c*), with MNLE being closer to reference performance than LAN[5,8,11] in 100, 91, and 66 out of 100 comparisons, respectively (all p < 0.01). Note that all three metrics were reported as averages over the four parameter dimensions of the DDM to keep the visualizations compact, and that this average did not affect the results qualitatively. We report metrics for each dimension in *Figure 3—figure supplement 1*, as well as additional inference accuracy results for smaller LAN simulation budgets (*Figure 3—figure supplement 2*) and for different numbers of observed trials (*Figure 3—figure supplement 3*).

Finally, we used the classifier 2-sample test (C2ST, *Lopez-Paz and Oquab, 2017*; *Lueckmann et al., 2021*) to quantify the similarity between the estimated and reference posterior distributions. The C2ST is defined to be the error rate of a classification algorithm which aims to classify whether samples belong to the true or the estimated posterior. Thus, it ranges from 0.5 (no difference between the distributions, the classifier is at chance level) to 1.0 (the classifier can perfectly distinguish the two

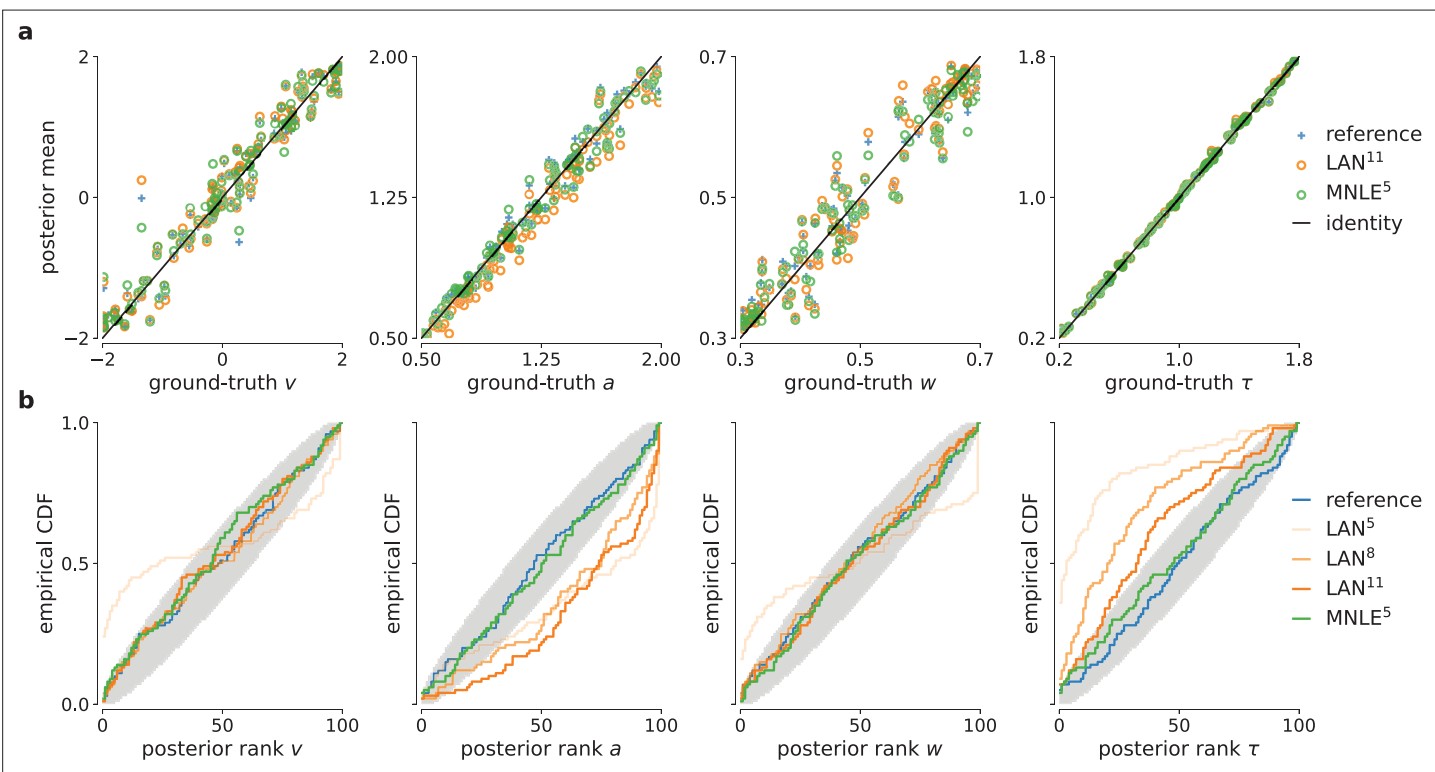

**Figure 4.** Parameter recovery and posterior uncertainty calibration for the drift-diffusion model (DDM). (**a**) Underlying ground-truth DDM parameters plotted against the sample mean of posterior samples inferred with the analytical likelihoods (reference, blue crosses), likelihood approximation network (LAN; orange circles), and mixed neural likelihood estimation (MNLE; green circles), for 100 different observations. Markers close to diagonal indicate good recovery of ground-truth parameters; circles on top of blue reference crosses indicate accurate posterior means. (**b**) Simulation-based calibration results showing empirical cumulative density functions (CDF) of the ground-truth parameters ranked under the inferred posteriors calculated from 100 different observations. A well-calibrated posterior must have uniformly distributed ranks, as indicated by the area shaded gray. Shown for reference posteriors (blue), LAN posteriors obtained with increasing simulation budgets (shades of orange, LAN[5,8,11]), and MNLE posterior (green, MNLE[5]), and for each parameter separately ($v$, $a$, $w$, and $\tau$).

The online version of this article includes the following figure supplement(s) for figure 4:

**Figure supplement 1.** Drift-diffusion model (DDM) parameter recovery for different numbers of observed trials.

**Figure supplement 2.** Drift-diffusion model (DDM) simulation-based calibration (SBC) results for different numbers of observed trials.

distributions). We note that the C2ST is a highly sensitive measure of discrepancy between two multi-variate distributions—for example if the two distributions differ in *any* dimension, the C2ST will be close to 1 even if all other dimensions match perfectly. We found that neither of the two approaches was able to achieve perfect approximations, but that MNLE[5] achieved lower C2ST scores compared to LAN[{5,8,11}] on all simulation budgets (*Figure 3e*): mean C2ST score LAN[{5,8,11}], 0.96, 0.78, 0.70 vs. MNLE[5], 0.65, with MNLE[5] showing a better score than LAN[{5,8,11}] on 100, 91, and 68 out of 100 pairwise comparisons, respectively (all p < 0.01). In summary, MNLE achieves more accurate recovery of posterior means than LANs, similar or better recovery of posterior variances, and overall more accurate posteriors (as quantified by C2ST).

## MNLE posteriors have uncertainties which are well calibrated

For practical applications of inference, it is often desirable to know how well an inference procedure can recover the ground-truth parameters, and whether the uncertainty estimates are well calibrated, (*Cook et al., 2006*), that is, that the *uncertainty* estimates of the posterior are balanced, and neither over-confident nor under-confident. For the DDM, we found that the posteriors inferred with MNLE and LANs (when using LAN[11]) recovered the ground-truth parameters accurately (in terms of posterior means, *Figure 3d* and *Figure 4a*)—in fact, parameter recovery was similarly accurate to using the 'true' analytical likelihoods, indicating that much of the residual error is due to stochasticity of the observations, and not the inaccuracy of the likelihood approximations.

To assess posterior calibration, we used simulation-based calibration (SBC, *Talts et al., 2018*). The basic idea of SBC is the following: If one repeats the inference with simulations from many different prior samples, then, with a well-calibrated inference method, the combined samples from all the inferred posteriors should be distributed according to the prior. One way to test this is to calculate the rank of each ground-truth parameter (samples from the prior) under its corresponding posterior, and to check whether all the ranks follow a uniform distribution. SBC results indicated that MNLE posteriors were as well calibrated as the reference posteriors, that is, the empirical cumulative density functions of the ranks were close to that of a uniform distribution (*Figure 4b*)—thus, on this example, MNLE inferences would likely be of similar quality compared to using the analytical likelihoods. When trained with the large simulation budget of $10^{11}$ simulations, LANs, too appeared to recover most of the ground-truth parameters well. However, SBC detected a systematic underestimation of the parameter *a* and overestimation of the parameter *τ*, and this bias increased for the smaller simulation budgets of LAN[5] and LAN[8] (*Figure 4b*, see the deviation below and above the desired uniform distribution of ranks, respectively).

The results so far (i.e., *Figures 3 and 4*) indicate that both LAN[11] and MNLE[5] lead to similar parameter recovery, but only MNLE[5] leads to posteriors which were well calibrated for all parameters. These results were obtained using a scenario with 100 i.i.d. trials. When increasing the number of trials (e.g., to 1000 trials), posteriors become very concentrated around the ground-truth value. In that case, while the posteriors overall identified the ground-truth parameter value very well (*Figure 4— figure supplement 1c*), even small deviations in the posteriors can have large effects on the posterior metrics (*Figure 3—figure supplement 3*). This effect was also detected by SBC, showing systematic biases for some parameters (*Figure 4—figure supplement 2*). For MNLE, we found that these biases were smaller, and furthermore that it was possible to mitigate this effect by inferring the posterior using ensembles, for example, by combining samples inferred with five MNLEs trained with identical settings but different random initialization (see Appendix 1 for details). These results show the utility of using SBC as a tool to test posterior coverage, especially when studying models for which reference posteriors are not available, as we demonstrate in the next section.

## MNLE infers well-calibrated, predictive posteriors for a DDM with collapsing bounds

MNLE was designed to be applicable to models for which evaluation of the likelihood is not practical so that standard inference tools cannot be used. To demonstrate this, we applied MNLE to a variant of the DDM for which analytical likelihoods are not available (note, however, that numerical approximation of likelihoods for this model would be possible, see e.g., *Shinn et al., 2020*, Materials and methods for details). This DDM variant simulates a decision variable like the simple DDM used above, but with linearly collapsing instead of constant decision boundaries (see e.g., *Hawkins et al.,*

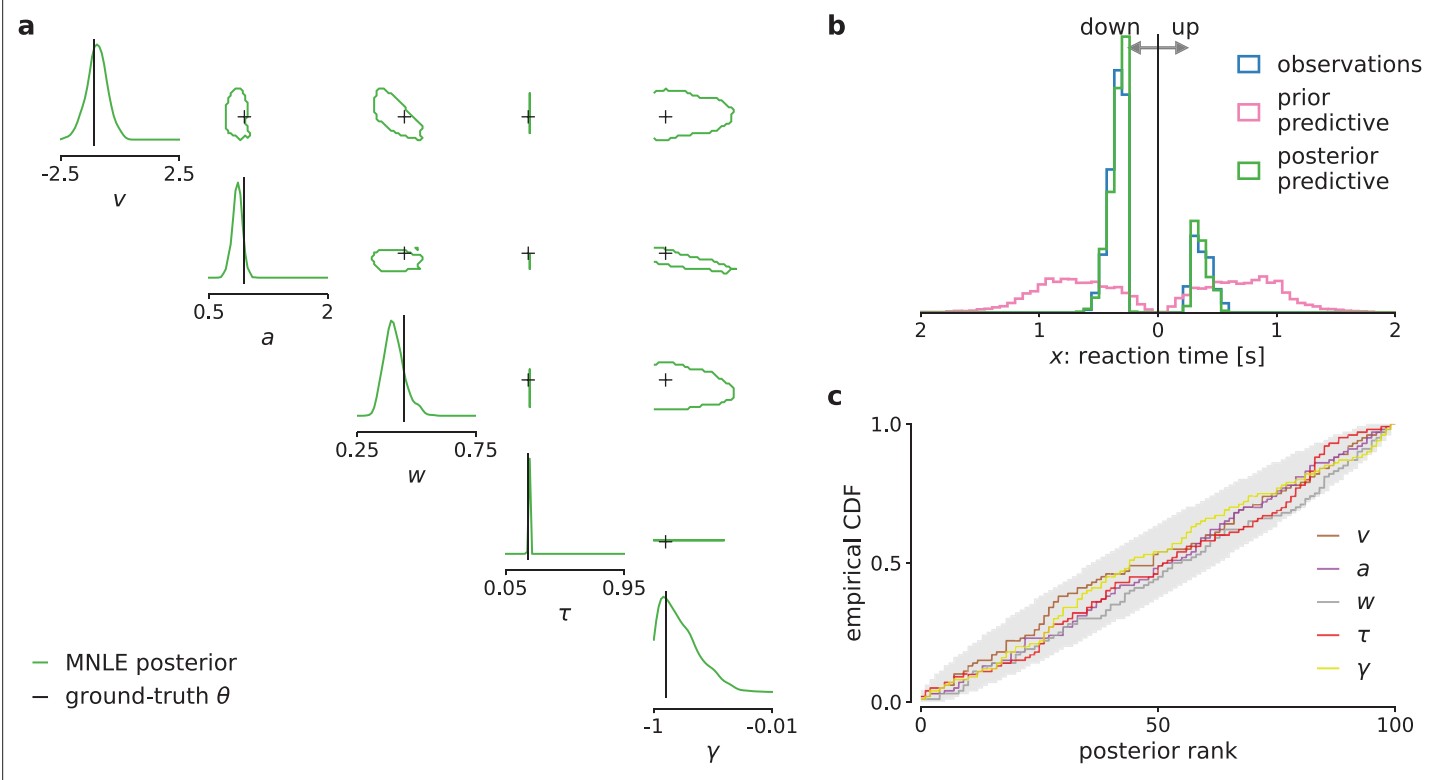

**Figure 5.** Mixed neural likelihood estimation (MNLE) infers accurate posteriors for the drift-diffusion model (DDM) with collapsing bounds. Posterior samples were obtained given 100-trial observations simulated from the DDM with linearly collapsing bounds, using MNLE and Markov Chain Monte Carlo (MCMC). (**a**) Approximate posteriors shown as 95% contour lines in a corner plot of one- (diagonal) and two-dimensional (upper triangle) marginals, for a representative 100-trial observation simulated from the DDM. (**b**) Reaction times and choices simulated from the ground-truth parameters (blue) compared to those simulated given parameters sampled from the prior (prior predictive distribution, purple) and from the MNLE posterior shown in (**a**) (posterior predictive distribution, green). (**c**) Simulation-based calibration results showing empirical cumulative density functions (CDF) of the ground-truth parameters ranked under the inferred posteriors, calculated from 100 different observations. A well-calibrated posterior must have uniformly distributed ranks, as indicated by the area shaded gray. Shown for each parameter separately ($v$, $a$, $w$, $\tau$, and $\gamma$).

*2015*; *Palestro et al., 2018*). The collapsing bounds are incorporated with an additional parameter $\gamma$ indicating the slope of the decision boundary, such that $\theta = (a, v, w, \tau, \gamma)$ (see Details of the numerical comparison).

We tested inference with MNLE on the DDM with linearly collapsing bound using observations comprised of 100 i.i.d. trials simulated with parameters sampled from the prior. Using the same MNLE training and MCMC settings as above, we found that posterior density concentrated around the underlying ground-truth parameters (see *Figure 5a*), suggesting that MNLE learned the underlying likelihood accurately. To assess inference quality systematically without needing reference posteriors, we performed posterior predictive checks by running simulations with the inferred posteriors samples and comparing them to observed (simulated) data, and checked posterior calibration properties using SBC. We found that the inferred posteriors have good predictive performance, that is, data simulated from the inferred posterior samples accurately matched the observed data (*Figure 5b*), and that their uncertainties are well calibrated as quantified by the SBC results (*Figure 5c*). Overall, this indicated that MNLE accurately inferred the posterior of this intractable variant of the DDM.

## Discussion

Statistical inference for computational models in cognitive neuroscience can be challenging because models often do not have tractable likelihood functions. The recently proposed LAN method (*Fengler et al., 2021*) performs SBI for a subset of such models (DDMs) by training neural networks with model simulations to approximate the intractable likelihood. However, LANs require large amounts of

training data, restricting its usage to canonical models. We proposed an alternative approached called MNLE, a synthetic neural likelihood method which is tailored to the data types encountered in many models of decision-making.

Our comparison on a tractable example problem used in *Fengler et al., 2021* showed that MNLE performed on par with LANs using six orders of magnitude fewer model simulations for training. While *Fengler et al., 2021* discuss that LANs were not optimized for simulation efficiency and that it might be possible to reduce the required model simulations, we emphasize that the difference in simulation efficiency is due to an inherent property of LANs. For each parameter in the training data, LANs require empirical likelihood targets that have to be estimated by building histograms or kernel density estimates from thousands of simulations. MNLE, instead, performs conditional density estimation without the need of likelihood targets and can work with only one simulation per parameter. Because of these conceptual differences, we expect the substantial performance advantage of MNLE to be robust to the specifics of the implementation.

After the networks are trained, the time needed for each evaluation determines the speed of inference. In that respect, both LANs and MNLEs are conceptually similar in that they require a single forward-pass through a neural network for each evaluation, and we found MNLE and the original implementation of LANs to require comparable computation times. However, evaluation time will depend, for example, on the exact network architecture, software framework, and computing infrastructure used. Code for a new PyTorch implementation of LANs has recently been released and improved upon the evaluation speed original implementation we compared to. While this new implementation made LAN significantly faster for a single forward-pass, we observed that the resulting time difference with the MCMC settings used here was only on the order of minutes, whereas the difference in simulation time for LAN[11] vs. MNLE[5] was on the order of days. The exact timings will always be implementation specific and whether or not these differences are important will depend on the application at hand. In a situation where iteration over model design is required (i.e., custom DDMs), an increase in training efficiency on the order of days would be advantageous.

There exist a number of approaches with corresponding software packages for estimating parameters of cognitive neuroscience models, and DDMs in particular. However, these approaches either only estimate single best-fitting parameters (*Voss and Voss, 2007*; *Wagenmakers et al., 2007*; *Chandrasekaran and Hawkins, 2019*; *Heathcote et al., 2019*; *Shinn et al., 2020*) or, if they perform fully Bayesian inference, can only produce Gaussian approximations to posteriors (*Feltgen and Daunizeau, 2021*), or are restricted to variants of the DDM for which the likelihood can be evaluated (*Wiecki et al., 2013*, HDDM [Hierarchical DDM] toolbox). A recent extension of the HDDM toolbox includes LANs, thereby combining HDDM's flexibility with LAN's ability to perform inference without access to the likelihood function (but this remains restricted to variants of the DDM for which LAN can be pretrained). In contrast, MNLE can be applied to any simulation-based model with intractable likelihoods and mixed data type outputs. Here, we focused on the direct comparison to LANs based on variants of the DDM. We note that these models have a rather low-dimensional observation structure (as common in many cognitive neuroscience models), and that our examples did not include additional parameter structure, for example, stimulus encoding parameters, which would increase the dimensionality of the learning problem. However, other variants of neural density estimation have been applied successfully to a variety of problems with higher dimensionality (see e.g., *Gonçalves et al., 2020*; *Lueckmann et al., 2021*; *Glöckler et al., 2021*; *Dax et al., 2022*). Therefore, we expect MNLE to be applicable to other simulation-based problems with higher-dimensional observation structure and parameter spaces, and to scale more favorably than LANs. Like for any neural network-based approach, applying MNLE to different inference problems may require selecting different architecture and training hyperparameters settings, which may induce additional computational training costs. To help with this process, we adopted default settings which have been shown to work well on a large range of SBI benchmarking problems (*Lueckmann et al., 2021*), and we integrated MNLE into the established sbi python package that provides well-documented implementations for training- and inference performance of SBI algorithms.

Several extensions to classical SL approaches have addressed the problem of a bias in the likelihood approximation due to the strong parametric assumptions, that is, Gaussianity, the use of summary statistics, or finite-sample biases (*Price et al., 2018*; *Ong et al., 2009*; *van Opheusden et al., 2020*). MNLE builds on flexible neural likelihood estimators, for example, normalizing flows, and does not

require summary statistics for a low-dimensional simulator like the DDM, so would be less susceptible to these first two biases. It could be subject to biases resulting from the estimation of the log-likelihoods from a finite number of simulations. In our numerical experiments, and for the simulation budgets we used, we did not observe biased inference results. We speculate that the ability of neural density estimators to pool data across multiple parameter settings (rather than using only data from a specific parameter set, like in classical SL methods) mitigates finite-sample effects.

MNLE is an SBI method which uses neural density estimators to estimate likelihoods. Alternatives to neural likelihood estimation include neural posterior estimation (NPE, *Papamakarios and Murray, 2016*; *Lueckmann et al., 2017*; *Greenberg et al., 2019*, which uses conditional density estimation to learn the posterior directly) and neural ratio estimation (NRE, *Hermans et al., 2020*; *Durkan et al., 2020*, which uses classification to approximate the likelihood-to-evidence ratio to then perform MCMC). These approaches could in principle be applied here as well, however, they are not as well suited for the flexible inference scenarios common in decision-making models as MNLE. NPE directly targets the posterior and therefore, by design, typically requires retraining if the structure of the problem changes (e.g., if the prior or the hierarchical structure of the model changes). There are variants of NPE that use embedding nets which can amortize over changing number of trials, avoiding retraining (*Radev et al., 2022*, *von Krause et al., 2022*). NRE learns the likelihood-to-evidence ratio using ratio estimation (and not density estimation) and would not provide an emulator of the simulator.

Regarding future research directions, MNLE has the potential to become more simulation efficient by using weight sharing between the discrete and the continuous neural density estimators (rather than to use separate neural networks, as we did here). Moreover, for high-dimensional inference problems in which slice sampling-based MCMC might struggle, MNLE could be used in conjunction with gradient-based MCMC methods like Hamiltonian Monte Carlo (HMC, *Brooks et al., 2011*; *Hoffman and Gelman, 2014*), or variational inference as recently proposed by *Wiqvist et al., 2021* and *Glöckler et al., 2021*. With MNLE's full integration into the sbi package, both gradient-based MCMC methods from Pyro (*Bingham et al., 2019*), and variational inference for SBI (SNVI, *Glöckler et al., 2021*) are available as inference methods for MNLE (a comparison of HMC and SNVI to slice sampling MCMC on one example observation resulted in indistinguishable posterior samples). Finally, using its emulator property (see Appendix 1), MNLE could be applied in an active learning setting for highly expensive simulators in which new simulations are chosen adaptively according to a acquisition function in a Bayesian optimization framework (*Gutmann and Corander, 2016*; *Lueckmann et al., 2019*; *Järvenpää et al., 2019*).

In summary, MNLE enables flexible and efficient inference of parameters of models in cognitive neuroscience with intractable likelihoods. The training efficiency and flexibility of the neural density estimators used overcome the limitations of LANs (*Fengler et al., 2021*). Critically, these features enable researchers to develop customized models of decision-making and not just apply existing models to new data. We implemented our approach as an extension to a public sbi python package with detailed documentation and examples to make it accessible for practitioners.

## Materials and methods
### Mixed neural likelihood estimation

MNLE extends the framework of neural likelihood estimation (*Papamakarios et al., 2019a*; *Lueckmann et al., 2019*) to be applicable to simulation-based models with mixed data types. It learns a parametric model $q_\psi(\mathbf{x}|\boldsymbol{\theta})$ of the intractable likelihood $p(\mathbf{x}|\boldsymbol{\theta})$ defined implicitly by the simulation-based model. The parameters $\psi$ are learned with training data $\{\boldsymbol{\theta}_n, \mathbf{x}_n\}_{1:N}$ comprised of model parameters $\boldsymbol{\theta}_n$ and their corresponding data simulated from the model $\mathbf{x}_n|\boldsymbol{\theta}_n \sim p(\mathbf{x}|\boldsymbol{\theta}_n)$. The parameters are sampled from a proposal distribution over parameters $\boldsymbol{\theta}_n \sim p(\boldsymbol{\theta})$. The proposal distribution could be any distribution, but it determines the parameter regions for which the density estimator will be good in estimating likelihoods. Thus, the prior, or a distribution that contains the support of the prior (or even all priors which one expects to use in the future) constitutes a useful choice. After training, the emulator can be used to generate synthetic data $\mathbf{x}|\boldsymbol{\theta} \sim q_\psi(\mathbf{x}|\boldsymbol{\theta})$ given parameters, and to evaluate the SL $q_\psi(\mathbf{x}|\boldsymbol{\theta})$ given experimentally observed data. Finally, the SL can be used to obtain posterior samples via

$$p(\boldsymbol{\theta}|\mathbf{x}) \propto q_\psi(\mathbf{x}|\boldsymbol{\theta})p(\boldsymbol{\theta}), \tag{1}$$

through approximate inference with MCMC. Importantly, the training is amortized, that is, the emulator can be applied to new experimental data without retraining (running MCMC is still required).

We tailored MNLE to simulation-based models that return mixed data, for example, in form of reaction times $rt$ and (usually categorical) choices $c$ as for the DDM. Conditional density estimation with normalizing flows has been proposed for continuous random variables (*Papamakarios et al., 2019a*), or discrete random variables (*Tran et al., 2019*), but not for mixed data. Our solution for estimating the likelihood of mixed data is to simply factorize the likelihood into continuous and discrete variables,

$$p(rt, c|\boldsymbol{\theta}) = p(rt|\boldsymbol{\theta}, c) \, p(c|\boldsymbol{\theta}), \tag{2}$$

and use two separate neural likelihood estimators: A discrete one $q_{\psi_c}$ to estimate $p(c|\boldsymbol{\theta})$ and a continuous one $q_{\psi_{rt}}$ to estimate $p(rt|\boldsymbol{\theta}, c)$. We defined $q_{\psi_c}$ to be a Bernoulli model and use a neural network to learn the Bernoulli probability $\rho$ given parameters $\boldsymbol{\theta}$. For $q_{\psi_{rt}}$ we used a conditional neural spline flow (NSF, *Durkan et al., 2019*) to learn the density of $rt$ given a parameter $\boldsymbol{\theta}$ and choice $c$. The two estimators are trained separately using the same training data (see Neural network architecture, training and hyperparameters for details). After training, the full neural likelihood can be constructed by multiplying the likelihood estimates $q_{\psi_c}$ and $q_{\psi_{rt}}$ back together:

$$q_{\psi_c, \psi_{rt}}(rt, c|\boldsymbol{\theta}) = q_{\psi_c}(c|\boldsymbol{\theta}) \, q_{\psi_{rt}}(rt|c, \boldsymbol{\theta}). \tag{3}$$

Note that, as the second estimator $q_{\psi_{rt}}(rt|c, \boldsymbol{\theta})$ is conditioned on the choice $c$, our likelihood model can account for statistical dependencies between choices and reaction times. The neural likelihood can then be used to approximate the intractable likelihood defined by the simulator, for example, for inference with MCMC. Additionally, it can be used to sample synthetic data given model parameters, without running the simulator (see The emulator property of MNLE).

## Relation to LAN

Neural likelihood estimation can be much more simulation efficient than previous approaches because it exploits continuity across the parameters by making the density estimation conditional. *Fengler et al., 2021*, too, aim to exploit continuity across the parameter space by training a neural network to predict the value of the likelihood function from parameters $\boldsymbol{\theta}$ and data $\mathbf{x}$. However, the difference to neural likelihood estimation is that they do not use the neural network for density estimation directly, but instead do classical neural network-based regression on likelihood targets. Crucially, the likelihood targets first have to obtained for each parameter in the training dataset. To do so, one has to perform density estimation using KDE (as proposed by *Turner et al., 2015*) or empirical histograms for *every* parameter separately. Once trained, LANs do indeed exploit the continuity across the parameter space (they can predict log-likelihoods given unseen data and parameters), however, they do so at a very high simulation cost: For a training dataset of $N$ parameters, they perform $N$ times KDE based on $M$ simulations each[11][1] For models with categorical output, that is, all decision-making models, KDE is performed separately for each choice., resulting is an overall simulation budget of $N \cdot M$ ($N = 1.5$ million and $M = 100,000$ for 'pointwise' LAN approach).

## Details of the numerical comparison

The comparison between MNLE and LAN is based on the DDM. The DDM simulates a decision variable $X$ as a stochastic differential equation with parameters $\boldsymbol{\theta} = (v, a, w, \tau)$:

$$dX_{t+\tau} = vdt + dW, \quad X_\tau = w, \tag{4}$$

where $W$ a Wiener noise process. The priors over the parameters are defined to be uniform: $v \sim \mathcal{U}(-2, 2)$ is the drift, $a \sim \mathcal{U}(0.5, 2)$ the boundary separation, $w \sim \mathcal{U}(0.3, 0.7)$ the initial offset, and $\tau \sim \mathcal{U}(0.2, 1.8)$ the nondecision time. A single simulation from the model returns a choice $c \in \{0, 1\}$ and the corresponding reaction time in seconds $rt \in (\tau, \infty)$.

For this version of the DDM the likelihood of an observation $(c, rt)$ given parameters $\boldsymbol{\theta}$, $L(c, rt|\boldsymbol{\theta})$, can be calculated analytically (*Navarro and Fuss, 2009*). To simulate the DDM and calculate analytical likelihoods we used the approach and the implementation proposed by *Drugowitsch, 2016*. We

numerically confirmed that the simulations and the analytical likelihoods match those obtained from the research code provided by *Fengler et al., 2021*.

To run LANs, we downloaded the neural network weights of the pretrained models from the repository mentioned in *Fengler et al., 2021*. The budget of training simulations used for the LANs was $1.5 \times 10^{11}$ (1.5 million training data points, each obtained from KDE based on $10^5$ simulations). We only considered the approach based on training a multilayer perceptron on single-trial likelihoods ('pointwise approach', *Fengler et al., 2021*). At a later stage of our study we performed additional experiments to evaluate the performance of LANs trained at smaller simulation budgets, for example, for $10^5$ and $10^8$ simulations. For this analysis, we used an updated implementation of LANs based on PyTorch that was provided by the authors. We used the training routines and default settings provided with that implementation. To train LANs at the smaller budgets we used the following splits of budgets into number of parameter settings drawn from the prior, and number of simulations per parameter setting used for fitting the KDE: for the $10^5$ budget we used $10^2$ and $10^3$, respectively (we ran experiments splitting the other way around, but results were slightly better for this split); for the $10^8$ budget we used an equal split of $10^4$ each (all code publicly available, see Code availability).

To run MNLE, we extended the implementation of neural likelihood estimation in the sbi toolbox (*Tejero-Cantero et al., 2020*). All comparisons were performed on a single AMD Ryzen Threadripper 1920X 12-Core processor with 2.2 GHz and the code is publicly available (see Code availability).

For the DDM variant with linearly collapsing decision boundaries, the boundaries were parametrized by the initial boundary separation, $a$, and one additional parameter, $\gamma$, indicating the slope with which the boundary approaches zero. This resulted in a five-dimensional parameter space for which we used the same prior as above, plus the an additional uniform prior for the slope $\gamma \sim \mathcal{U}(-1.0, 0)$. To simulate this DDM variant, we again used the Julia package by *Drugowitsch, 2016*, but we note that for this variant no analytical likelihoods are available. While it would be possible to approximate the likelihoods numerically using the Fokker–Planck equations (see e.g., *Shinn et al., 2020*), this would usually involve a trade-off between computation time and accuracy as well as design of bespoke solutions for individual models, and was not pursued here.

## Flexible Bayesian inference with MCMC

Once the MNLE is trained, it can be used for MCMC to obtain posterior samples $\boldsymbol{\theta} \sim p(\boldsymbol{\theta}|\mathbf{x})$ given experimentally observed data $\mathbf{x}$. To sample from posteriors via MCMC, we transformed the parameters to unconstrained space, used slice sampling (*Neal, 2009*), and initialized ten parallel chains using sequential importance sampling (*Papamakarios et al., 2019a*), all as implemented in the sbi toolbox. We ran MCMC with identical settings for MNLE and LAN.

Importantly, performing MNLE and then using MCMC to obtain posterior samples allows for flexible inference scenarios because the form of $\mathbf{x}$ is not fixed. For example, when the model produces trial-based data that satisfy the i.i.d. assumption, for example, a set of reaction times and choices $\mathbf{X} = \{rt, c\}_{i=1}^{N}$ in a DDM, then MNLE allows to perform inference given varying numbers of trials, without retraining. This is achieved by training MNLE based on single-trial likelihoods once and then combining multiple trials into the joint likelihood only when running MCMC:

$$p(\boldsymbol{\theta}|\mathbf{X}) \propto \prod_{i=1}^{N} q(rt_i, c_i|\boldsymbol{\theta}) \, p(\boldsymbol{\theta}). \tag{5}$$

Similarly, one can use the neural likelihood to perform hierarchical inference with MCMC, all without the need for retraining (see *Hermans et al., 2020*; *Fengler et al., 2021*, for examples).

## Stimulus- and intertrial dependencies

Simulation-based models in cognitive neuroscience often depend not only on a set of parameters $\boldsymbol{\theta}$, but additionally on (a set of) stimulus variables $s$ which are typically given as part of the experimental conditions. MNLE can be readily adapted to this scenario by generating simulated data with multiple stimulus variables, and including them as additional inputs to the network during inference. Similarly, MNLE could be adapted to scenarios in which the i.i.d. assumption across trials as used above (see Flexible Bayesian inference with MCMC) does not hold. Again, this would be achieved by adapting the model simulator accordingly. For example, when inferring parameters $\boldsymbol{\theta}$ of a DDM for

which the outcome of the current trial $i$ additionally depends on current stimulus variables $s_i$ as well as on previous stimuli $s_j$ and responses $r_j$, then one would implement the DDM simulator as a function $f(\boldsymbol{\theta}; s_{i-T}, \ldots, s_i; r_{i-T}, \ldots, r_{i-1})$ (where $T$ is a history parameter) and then learn the underlying likelihood by emulating $f$ with MNLE.

## Neural network architecture, training, and hyperparameters

### Architecture

For the architecture of the Bernoulli model we chose a feed-forward neural network that takes parameters $\boldsymbol{\theta}$ as input and predicts the Bernoulli probability $\rho$ of the corresponding choices. For the normalizing flow we used the NSF architecture (*Durkan et al., 2019*). NSFs use a standard normal base distribution and transform it using several modules of monotonic rational-quadratic splines whose parameters are learned by invertible neural networks. Using an unbounded base distribution for modeling data with bounded support, for example, reaction time data $rt \in (0, \infty)$, can be challenging. To account for this, we tested two approaches: We either transformed the reaction time data to logarithmic space to obtain an unbounded support ($\log rt \in (-\infty, \infty)$), or we used a log-normal base distribution with rectified (instead of linear) tails for the splines (see *Durkan et al., 2019*, for details and Architecture and training hyperparameters for the architecture settings used).

### Training

The neural network parameters $\psi_c$ and $\psi_{rt}$ were trained using the maximum likelihood loss and the Adam optimizer (*Kingma and Ba, 2015*). As proposal distribution for the training dataset we used the prior over DDM parameters. Given a training dataset of parameters, choices, and reaction times $\{\boldsymbol{\theta}_i, (c_i, rt_i)\}_{i=1}^N$ with $\boldsymbol{\theta}_i \sim p(\boldsymbol{\theta})$; $(c_i, rt_i) \sim \mathrm{DDM}(\boldsymbol{\theta}_i)$, we minimized the negative log-probability of the model. In particular, using the same training data, we trained the Bernoulli choice model by minimizing

$$-\frac{1}{N} \sum_{i=1}^N \log q_{\psi_c}(c_i | \boldsymbol{\theta}_i), \tag{6}$$

and the NSF by minimizing

$$-\frac{1}{N} \sum_{i=1}^N \log q_{\psi_{rt}}(rt | c_i, \boldsymbol{\theta}_i). \tag{7}$$

Training was performed with code and training hyperparameter settings provided in the sbi toolbox (*Tejero-Cantero et al., 2020*).

### Hyperparameters

MNLE requires a number of hyperparameter choices regarding the neural network architectures, for example, number of hidden layers, number of hidden units, number of stacked NSF transforms, kind of base distribution, among others (*Durkan et al., 2019*). With our implementation building on the sbi package we based our hyperparameter choices on the default settings provided there. This resulted in likelihood accuracy similar to LAN, but longer evaluation times due to the complexity of the underlying normalizing flow architecture.

To reduce evaluation time of MNLE, we further adapted the architecture to the example model (DDM). In particular, we ran a cross-validation of the hyperparameters relevant for evaluation time, that is, number of hidden layers, hidden units, NSF transforms, spline bins, and selected those that were optimal in terms of Huber loss and MSE between the approximate and the *analytical* likelihoods, as well as evaluation time. This resulted in an architecture with performance *and* evaluation time similar to LANs (more details in Appendix: Architecture and training hyperparameters). The cross-validation relied on access to the analytical likelihoods which is usually not given in practice, for example, for simulators with intractable likelihoods. However, we note that in cases without access to analytical likelihoods a similar cross-validation can be performed using quality measures other than the difference to the analytical likelihood, for example, by comparing the observed data with synthetic data and SLs provided by MNLE.

## Acknowledgements

We thank Luigi Acerbi, Michael Deistler, Alexander Fengler, Michael Frank, and Ingeborg Wenger for discussions and comments on a preliminary version of the manuscript. We also acknowledge and thank the Python and Julia communities for developing the tools enabling this work, including Diff erentialEquations.jl, DiffModels.jl, NumPy, pandas, Pyro, PyTorch, sbi, sbibm, and Scikit-learn (see Appendix for details).

## Additional information

### Funding

| Funder | Grant reference number | Author |
| --- | --- | --- |
| Deutsche Forschungsgemeinschaft | SFB 1233 | Jan-Matthis Lueckmann Jakob H Macke |
| Deutsche Forschungsgemeinschaft | SPP 2041 | Jan Boelts Jakob H Macke |
| Deutsche Forschungsgemeinschaft | Germany's Excellence Strategy MLCoE | Jan Boelts Jan-Matthis Lueckmann Richard Gao Jakob H Macke |
| Bundesministerium für Bildung und Forschung | ADIMEM | Jan-Matthis Lueckmann Jakob H Macke |
| HORIZON EUROPE Marie Sklodowska-Curie Actions | 101030918 | Richard Gao |
| Bundesministerium für Bildung und Forschung | Tübingen AI Center | Jan Boelts Jakob H Macke |
| Bundesministerium für Bildung und Forschung | FKZ 01IS18052 A-D | Jan-Matthis Lueckmann Jakob H Macke |
| Bundesministerium für Bildung und Forschung | KZ 01IS18039A | Jan Boelts Jakob H Macke |

The funders had no role in study design, data collection, and interpretation, or the decision to submit the work for publication.

### Author contributions

Jan Boelts, Conceptualization, Resources, Data curation, Software, Formal analysis, Validation, Investigation, Visualization, Methodology, Writing – original draft, Writing – review and editing; Jan-Matthis Lueckmann, Conceptualization, Supervision, Visualization, Methodology, Writing – original draft, Writing – review and editing; Richard Gao, Conceptualization, Software, Validation, Visualization, Writing – original draft, Writing – review and editing; Jakob H Macke, Conceptualization, Supervision, Funding acquisition, Methodology, Project administration, Writing – review and editing

### Author ORCIDs

Jan Boelts (iD) http://orcid.org/0000-0003-4979-7092
Richard Gao (iD) http://orcid.org/0000-0001-5916-6433
Jakob H Macke (iD) http://orcid.org/0000-0001-5154-8912

### Decision letter and Author response

Decision letter https://doi.org/10.7554/eLife.77220.sa1
Author response https://doi.org/10.7554/eLife.77220.sa2

## Additional files

### Supplementary files
• Transparent reporting form

## Data availability

We implemented MNLE as part of the open source package for SBI, sbi, available at https://github.com/mackelab/sbi, copy archived at swh:1:rev:d72fc6d790285c7779afbbe9a5f6b640691d4560. Code for reproducing the results presented here, and tutorials on how to apply MNLE to other simulators using sbi can be found at https://github.com/mackelab/mnle-for-ddms, copy archived at swh:1:rev:5e6cf714c223ec5c414b76ac70f7dc88d4fbd321.

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

# Appendix 1

## Code availability

We implemented MNLE as part of the open-source package for SBI, sbi, available at https://github.com/mackelab/sbi, (*Boelts, 2022a* copy archived at swh:1:rev:d72fc6d790285c7779afbbe9a5f6b640691d4560). Code for reproducing the results presented here, and tutorials on how to apply MNLE to other simulators using sbi can be found at https://github.com/mackelab/mnle-for-ddms, (*Boelts, 2022b* copy archived at swh:1:rev:5e6cf714c223ec5c414b76ac70f7dc88d4fbd321). The implementation of MNLE relies on packages developed by the Python (*Van Rossum and Drake, 1995*) and Julia (*Bezanson et al., 2017*) communities, including DifferentialEquations.jl (*Rackauckas and Nie, 2017*), DiffModels.jl (*Drugowitsch, 2016*), NumPy (*Harris et al., 2020*), pandas (*pandas development team, 2020*), Pyro (*Bingham et al., 2019*), PyTorch (*Paszke et al., 2019*), sbi (*Tejero-Cantero et al., 2020*), sbibm (*Lueckmann et al., 2021*), and Scikit-learn (*Pedregosa et al., 2011*).

## Architecture and training hyperparameters

For the Bernoulli neural network we used three hidden layers with 10 units each and sigmoid activation functions. For the neural spline flow architecture (*Durkan et al., 2019*), we transformed the reaction time data to the log-domain, used a standard normal base distribution, 2 spline transforms with 5 bins each and conditioning networks with 3 hidden layers and 10 hidden units each, and rectified linear unit activation functions. The neural network training was performed using the sbi package with the following settings: learning rate 0.0005; training batch size 100; 10% of training data as validation data, stop training after 20 epochs without validation loss improvement.

## The emulator property of MNLE

Being based on the neural likelihood estimation framework, MNLE naturally returns an emulator of the simulator that can be sampled to generate synthetic data without running the simulator. We found that the synthetic data generated by MNLE accurately matched the data we obtained by running the DDM simulator (*Figure 2—figure supplement 1*). This has several potential benefits: it can help with evaluating the performance of the density estimator, it enables almost instantaneous data generation (one forward-pass in the neural network) even if the simulator is computationally expensive, and it gives full access to the internals of the emulator, for example, to gradients w.r.t. to data or parameters.

There is variant of the LAN approach which allows for sampling synthetic data as well: In the 'Histogram-approach' (*Fengler et al., 2021*) LANs are trained with a convolutional neural network (CNN) architecture using likelihood targets in form of two-dimensional empirical histograms. The output of the CNN is a probability distribution over a discretized version of the data space which can, in principle, be sampled to generate synthetic DDM choices and reaction times. However, the accuracy of this emulator property of CNN-LANs is limited by the number of bins used to approximate the continuous data space (e.g., 512 bins for the examples shown in *Fengler et al., 2021*).

