## [Editor Report]

This paper provides a new approach, Mixed Neural Likelihood Estimator (MNLE) to build likelihood emulators for simulation-based models where the likelihood is unavailable. The authors show that the MNLE approach is equally accurate but orders of magnitude more efficient than a recent proposal, likelihood approximation networks (LAN), on two variants of the drift-diffusion model (a widely used model in cognitive neuroscience). This work provides a practical approach for fitting more complex models of behavior and neural activity for which likelihoods are unavailable.

---

## [Decision Letter]

**Decision letter after peer review:**

Thank you for submitting your article "Flexible and efficient simulation-based inference for models of decision-making" for consideration by *eLife*. Your article has been reviewed by 2 peer reviewers, and the evaluation has been overseen by Valentin Wyart as the Reviewing Editor and Timothy Behrens as the Senior Editor. The following individuals involved in review of your submission have agreed to reveal their identity: Luigi Acerbi (Reviewer #1); Jean Daunizeau (Reviewer #2).

The reviewers have discussed their reviews with one another, and the Reviewing Editor has drafted this to help you prepare a revised submission. As you will see after reading the two reviews (copied at the bottom of this message), the two reviewers have found that the MNLE approach you describe represents a significant advance over LANs. As summarized by Reviewer #1, instead of estimating the likelihood *independently* for each parameter setting and then training a neural network to interpolate between these density estimates, MNLE uses *conditional* density estimation which trains a density network while sharing information across different parameters settings. This approach, and others built on top of it or inspired by it, may have a large impact in the field, even more so since all code has been made available by the authors.

But while this approach is a priori order of magnitude more efficient than LANs, the reviewers have agreed that the current manuscript does not provide sufficient empirical evidence that it is the case. The reviewers have discussed how in practice the comparison between LANs and MNLEs could be improved to strengthen the manuscript in this respect. Furthermore, the reviewers have also identified limitations of the approach (shared with LANs) that are worth discussing more explicitly in the paper. We hope that you will find the reviews helpful when revising your manuscript. When submitting your revised manuscript, please provide a point-by-point response to the essential revisions that were decided after discussion between the reviewers. The other points made in the separate reviews (provided at the bottom of this message) should also be addressed, but they do not require a point-by-point response.

Essential revisions:

1. While the MNLE approach appears much more sample-efficient than the LAN approach, there is not sufficient empirical evidence that it is indeed the case. The main piece of evidence is described in Section 2.2, where the authors show that MNLE learns likelihood function as accurately as LAN, but with only a fraction of the simulation budget. The authors show that a MNLE estimator trained with 10^5 model simulations is as accurate as a LAN estimator trained with 10^11 model simulations. However, the authors did not show that a LAN estimator trained with only 10^5 model simulations yields less accurate results than a MNLE estimator trained with the same simulation budget, or that a MNLE estimator trained with 10^11 simulations yields much better results than a LAN estimator trained with the same simulation budget. A more interpretable comparison of the two approaches is needed here: claiming that MNLE is more efficient than LAN is important, but requires additional work. We do not require the authors to systematically vary the number of model simulations (spanning the 10^5 to 10^11 range), and quantify each method's efficiency in terms of accuracy profile over this range. The authors could compare their MNLE estimator trained with 10^5 model simulations with a LAN estimator trained on 10^8 samples, which is still 1000 times more than their MNLE estimator, and show that the LAN estimator performs much worse than their MNLE estimator. Note that there is an additional technical caveat here, in that training LANs depends on an additional degree of freedom = how many parameters / how many samples per parameter. The authors should keep a sensible split. The current split appears to be 10^6 parameters and 10^5 samples per parameter. When going down to 10^8, it would make sense to keep a similar ratio.

2. Additional "accuracy" metrics should be considered when comparing MNLE and LAN estimators. These metrics may include likelihood approximation accuracy, parameter estimation accuracy and estimation uncertainty accuracy. The last two are important in practice, because errors in likelihood approximations may have a very small impact on parameter estimation. Regarding parameter estimation accuracy, the metric used in the manuscript (the "absolute difference in posterior sample mean normalized by reference posterior standard deviation") may not be the most informative. Indeed, significant differences between MNLEs and LANs in this metric may have no practical consequence whatsoever in terms of absolute parameter estimation errors. This is what Figure 4A seems to indicate. It could be useful to use metrics that relate to parameter recovery, e.g., parameter estimation error or pairwise parameter confusion. LAN estimators may show more pairwise parameter confusion than MNLE estimators, for example.

3. The following limitations of MNLE estimators (limitations shared with LAN estimators) should be discussed more explicitly in the manuscript:

– The drift-diffusion model used to benchmark the approach has a very low-dimensional observation structure (one binary observation + one continuous observation per trial). This limitation is not necessarily a problem because many models in cognitive neuroscience have a very low-dimensional observation structure, but it is worth mentioning.

– The approach described works for i.i.d. data. Any additional structure/dependence (e.g., adding parameters to characterize the stimulus shown in the trial) effectively increases the dimensionality of the likelihood approximation the network needs to learn.

– The manuscript does not sufficiently discuss nor explore the scalability of the MNLE approach. Given the previous related work by some of the authors, that has shown remarkable results in this respect, it would be useful to discuss the scalability of MNLEs in the discussion.

– Like any neural-network based approach, selecting the hyperparameters and architecture of the network requires either prior knowledge and/or brute force. And it is currently unclear how in practice the MNLE approach can be applied for different models. Importantly, even if training a MNLE estimator *given the hyperparameters* might require a small number of simulations (10^5), we need to account for the additional cost of finding the correct hyperparameters for training the emulator (presumably, requiring an exploration of at least 10-100 hyperparameter settings).

*Reviewer #1 (Recommendations for the authors):*

I already discussed an earlier version of the manuscript with the authors, so I don't really have much to add to my previous comments and to my public comments (i.e., there are a couple of limitations that the authors could address a bit more explicitly, like dimensionality of the method and the cost of hyperparameter tuning).

*Reviewer #2 (Recommendations for the authors):*

Let me now explain why I think that authors' main claim (namely: that MNLE is more efficient than LAN) may not be strongly supported by the results reported in the current version of the manuscript.

The main piece of evidence is summarized in Section 2.2, where authors claim that MNLE learns likelihood function as accurately as LAN, but with only a fraction of the simulation budget. In brief, authors showed that a MNLE estimator trained with 10^5 model simulations is as accurate as a pre-trained version of the LAN method for DDMs (which used 10^11 model simulations). The problem here, is that they did not show that a LAN estimator trained with only 10^5 model simulations yields less accurate results. Or that MNLE trained with 10^11 simulations yields much better results. More generally, what is needed here is a quantitative comparison of the efficiency of the method. One would expect that, for both MNLE and LAN, increasing the number of model simulations increases the accuracy of likelihood approximation. However, this probably follows the law of diminishing marginal utility (i.e. using 10^11 simulations may bring little advantage when compared to e.g. 10^10) (:) Hence, efficiency here should be measured in terms of the speed at which the results accuracy increases. In other terms, claiming that MNLE is more efficient than LAN is important, but requires more work than what is done here. Authors should systematically vary the number of model simulations (spanning, e.g. the 10^5 to 10^11 range), and quantify each method's efficiency in terms of the accuracy profile over this range.

Now, this analysis should be performed with different "accuracy" metrics. In particular, authors may measure likelihood approximation accuracy, parameter estimation accuracy and estimation uncertainty accuracy. The latter are critical, because if errors in likelihood approximations may have very small impact on parameter estimation. I note that, w.r.t. parameter estimation accuracy and estimation uncertainty accuracy, I don't think authors chose very informative metrics. For example, the former is defined as the "absolute difference in posterior sample mean normalized by reference posterior standard deviation". If I understood correctly, this metric may show significant differences between MNLE and LAN that would have no practical consequence whatsoever, given that methods would be very similar in terms of their absolute parameter estimation errors. In fact, this is what Figure 4A seems to indicate. I would rather suggest to use accuracy metrics that practically "mean something" for parameter recovery. Measuring parameter estimation error is a possibility (although one may then conclude that MNLE brings no clear advantage when compared to LAN, cf. Figure 4A). But there are other possibilities. For example, likelihood approximation errors may induce pairwise parameter confusions. In turn, for a small simulation budget, LAN estimators may show more pairwise parameter confusion than MNLE. Note: to quantify pairwise parameter confusion in the context of DDM, authors may take inspiration from Feltgen 2021 (or any other metric of their choice).

---

## [Author Response]

The reviewers have discussed their reviews with one another, and the Reviewing Editor has drafted this to help you prepare a revised submission. As you will see after reading the two reviews (copied at the bottom of this message), the two reviewers have found that the MNLE approach you describe represents a significant advance over LANs. As summarized by Reviewer #1, instead of estimating the likelihood *independently* for each parameter setting and then training a neural network to interpolate between these density estimates, MNLE uses *conditional* density estimation which trains a density network while sharing information across different parameters settings. This approach, and others built on top of it or inspired by it, may have a large impact in the field, even more so since all code has been made available by the authors.But while this approach is a priori order of magnitude more efficient than LANs, the reviewers have agreed that the current manuscript does not provide sufficient empirical evidence that it is the case. The reviewers have discussed how in practice the comparison between LANs and MNLEs could be improved to strengthen the manuscript in this respect. Furthermore, the reviewers have also identified limitations of the approach (shared with LANs) that are worth discussing more explicitly in the paper. We hope that you will find the reviews helpful when revising your manuscript. When submitting your revised manuscript, please provide a point-by-point response to the essential revisions that were decided after discussion between the reviewers. The other points made in the separate reviews (provided at the bottom of this message) should also be addressed, but they do not require a point-by-point response.

We appreciate that the reviewers view our approach as a significant advance over LANs, and with a potentially large impact on the field. We agree that, in the previous version of the manuscript, there was a lack of a direct comparison between MNLE and LANs at a similar simulation budget. We now provide results for these experiments below and show that MNLE is significantly more accurate than LANs across several simulation budgets. We now also provide further discussions regarding limitations of the MNLE.

Essential revisions:1. While the MNLE approach appears much more sample-efficient than the LAN approach, there is not sufficient empirical evidence that it is indeed the case. The main piece of evidence is described in Section 2.2, where the authors show that MNLE learns likelihood function as accurately as LAN, but with only a fraction of the simulation budget. The authors show that a MNLE estimator trained with 10^5 model simulations is as accurate as a LAN estimator trained with 10^11 model simulations. However, the authors did not show that a LAN estimator trained with only 10^5 model simulations yields less accurate results than a MNLE estimator trained with the same simulation budget, or that a MNLE estimator trained with 10^11 simulations yields much better results than a LAN estimator trained with the same simulation budget. A more interpretable comparison of the two approaches is needed here: claiming that MNLE is more efficient than LAN is important, but requires additional work. We do not require the authors to systematically vary the number of model simulations (spanning the 10^5 to 10^11 range), and quantify each method's efficiency in terms of accuracy profile over this range. The authors could compare their MNLE estimator trained with 10^5 model simulations with a LAN estimator trained on 10^8 samples, which is still 1000 times more than their MNLE estimator, and show that the LAN estimator performs much worse than their MNLE estimator. Note that there is an additional technical caveat here, in that training LANs depends on an additional degree of freedom = how many parameters / how many samples per parameter. The authors should keep a sensible split. The current split appears to be 10^6 parameters and 10^5 samples per parameter. When going down to 10^8, it would make sense to keep a similar ratio.

The revised manuscript includes additional experiments to show that MNLE is more simulation-efficient than LANs, as requested, with updated main figures. Note that the authors of the LAN paper recently provided an updated PyTorch implementation of LAN, which now made it feasible to retrain LANs at smaller budgets locally, instead of having to exclusively rely on the pre-trained network at a budget of 10^11^, as we had done previously.

We retrained LANs for budgets of 10^5^ and 10^8^ simulations. For the 10^5^ budget we used 10^2^ parameter samples and 10^3^ samples per parameter, respectively (in contrast to the split used in the LAN paper, we used more samples per parameter than parameter samples, which resulted in slightly better performance than the other way around). For the 10^8^ budget we used an equal split of 10^4^ and 10^4^.

As neural network-training settings, we used the defaults provided in the new LAN implementation. As training data, we used the data from the same DDM simulator as used for MNLE, except for the 10^11^ LAN budget for which simulation and training was computationally too expensive so that we used the updated pre-trained network (the training data still came from the same distribution (simple DDM), just not from our own simulator).

We found that MNLE with a 10^5^ budget (henceforth referred to as MNLE^5^) significantly outperformed LAN trained with 10^5^ and 10^8^ budgets (LAN^5^ and LAN^8^, respectively) on all metrics (see updated Figure 2 below). When compared to LAN^11^, MNLE^5^ was more accurate on all but one metric, the Huber loss on the log-likelihoods (LAN’s training loss). Note that the numbers for MNLE^5^ and LAN^11^ differ slightly from our initial submission because MNLE performance is now averaged across ten random neural network initializations (a subset of which outperformed the single pre-trained LAN^11^ network) and because we used new implementations of LAN (provided by the authors) and for MNLE (now fully integrated into the *sbi* toolbox).

We updated Figure 2 with the new results and the following changes: we added the new results based on the smaller simulation budgets for LANs as lighter shades of orange; we show the metrics results as bar plots with standard error of the mean (SEM) error bars instead of boxplots, to improve readability; we replace panels D (evaluation time) and E (budgets) with Huber loss and MSE calculated on log-likelihoods (shown before in panel B and C) and show Huber loss and MSE on calculated on likelihoods in panel B and C. The main text is updated accordingly (not copied here due to length). We also still mention the results on evaluation time in the main text, and cover inference time of LAN and MNLE in general in the discussion as well.

2. Additional "accuracy" metrics should be considered when comparing MNLE and LAN estimators. These metrics may include likelihood approximation accuracy, parameter estimation accuracy and estimation uncertainty accuracy. The last two are important in practice, because errors in likelihood approximations may have a very small impact on parameter estimation. Regarding parameter estimation accuracy, the metric used in the manuscript (the "absolute difference in posterior sample mean normalized by reference posterior standard deviation") may not be the most informative. Indeed, significant differences between MNLEs and LANs in this metric may have no practical consequence whatsoever in terms of absolute parameter estimation errors. This is what Figure 4A seems to indicate. It could be useful to use metrics that relate to parameter recovery, e.g., parameter estimation error or pairwise parameter confusion. LAN estimators may show more pairwise parameter confusion than MNLE estimators, for example.

We thank the reviewers for pointing this out and agree that it is important to check how errors in likelihood approximation accuracy impact inference accuracy in practice, and to report parameter estimation accuracy with a separate metric. To evaluate both posterior accuracy and parameter estimation accuracy we now additionally report the mean squared error (MSE) between the posterior sample mean and the underlying ground-truth parameters. We updated figure 3 as follows: in panel B and C we show relative errors in posterior mean and posterior variance (as before), and in panel D we now show the parameter estimation accuracy for all methods, as well as for the reference posterior as a reference for the best possible accuracy (the panel showed the dispersion metric before). Note that we average parameter estimation metric over the four parameter dimensions to keep the figure compact, but that this did not affect results qualitatively (we added a supplementary figure showing metrics for each parameter dimension separately, Figure 3—figure supplement 1, see end of document). Panel E shows C2ST scores as before. All panels were updated with the corresponding results for inference with LAN budgets 10^5^ and 10^8^ as well.

We found that the increased errors in likelihood approximation accuracies for smaller LAN budgets we reported above were reflected in the inference accuracy as well (see updated Figure 3):

To provide concrete examples of how using a LAN trained on a smaller budget affects the inferred posterior we added a supplementary figure (Figure 3—figure supplement 2, see end of document) showing posteriors inferred with the smaller LAN budgets. Additionally, we calculated simulation-based calibration (SBC) results for the two smaller LAN budgets and found that calibration quality decreased with simulation budget and updated Figure 4B, the figure caption and the main text accordingly.

We thank reviewer 2 for the pointer to the parameter confusion as an additional metric. However, we did not obtain this additional result as we believe that the current results on inference accuracy for the smaller simulation budgets convincingly demonstrate the efficiency of MNLE versus LAN. Overall, this shows that for budgets smaller than 10^11^ simulations, inference with LANs was not nearly as accurate as with MNLE, whereas MNLE was close to the reference performance on parameter estimation accuracy. We adapted the figure captions and results reported in the main text accordingly.

3. The following limitations of MNLE estimators (limitations shared with LAN estimators) should be discussed more explicitly in the manuscript:– The drift-diffusion model used to benchmark the approach has a very low-dimensional observation structure (one binary observation + one continuous observation per trial). This limitation is not necessarily a problem because many models in cognitive neuroscience have a very low-dimensional observation structure, but it is worth mentioning.– The approach described works for i.i.d. data. Any additional structure/dependence (e.g., adding parameters to characterize the stimulus shown in the trial) effectively increases the dimensionality of the likelihood approximation the network needs to learn.– The manuscript does not sufficiently discuss nor explore the scalability of the MNLE approach. Given the previous related work by some of the authors, that has shown remarkable results in this respect, it would be useful to discuss the scalability of MNLEs in the discussion.– Like any neural-network based approach, selecting the hyperparameters and architecture of the network requires either prior knowledge and/or brute force. And it is currently unclear how in practice the MNLE approach can be applied for different models. Importantly, even if training a MNLE estimator *given the hyperparameters* might require a small number of simulations (10^5), we need to account for the additional cost of finding the correct hyperparameters for training the emulator (presumably, requiring an exploration of at least 10-100 hyperparameter settings).

We agree that these are four important points and changed the third paragraph in the discussion accordingly (see below).

Regarding the additional model or stimulus parameter, we agree that those will effectively increase the dimensionality of the learning problem, but we note that under the i.i.d assumption, additional parameter dependencies, e.g,. for hierarchical inference scenarios, will not require retraining MNLEs but will only affect the inference step with MCMC or VI.

Regarding the selection of neural network settings we agree that this was not pointed out clearly enough in the manuscript. MNLE is now fully integrated into the *sbi* toolbox and was set up with default settings that we expect to work well in many applications (see e.g., the sbi benchmark by Lueckmann et al., 2021, who showed that the same settings worked well for NLE across a wide range of tasks). Additionally, with the evaluation techniques available in the *sbi* toolbox, e.g., posterior predictive checks and simulation-based calibration, it is now possible for users to evaluate their trained MNLE emulator. Nevertheless, it is important to emphasize that it might be necessary to adapt the MNLE architecture to a given application and to repeat training multiple times to try different settings.

To address all four points, we adapted the discussion as follows (extending the third paragraph in the discussion):

“[…]. In contrast, MNLE can be applied to any simulation-based model with intractable likelihoods and mixed data type-outputs. Here, we focused on the direct comparison to LANs based on variants of the DDM. We note that these models have a rather low-dimensional observation structure (as common in many cognitive neuroscience models), and that our examples did not include additional parameter structure, e.g., stimulus encoding parameters, which would increase the dimensionality of the learning problem. However, other variants of neural density estimation have been applied successfully to a variety of problems with higher dimensionality (see e.g., Gonçalves et al., 2020; Lueckmann et al., 2021; Glöckler et al., 2021; Dax et al., 2022). Therefore, we expect MNLE to be applicable to other simulation-based problems with higher-dimensional observation structure and parameter spaces, and to scale more favorably than LANs.

Like for any neural network-based approach, applying MNLE to different inference problems may require selecting different architecture and training hyperparameters settings, which may induce additional computational training costs. To help with this process, we adopted default settings which have been shown to work well on a large range of SBI benchmarking problems (Lueckmann et al., 2021), and we integrated MNLE into the established sbi python package that provides well-documented implementations for training- and inference performance of SBI algorithms.”

Reviewer #2 (Recommendations for the authors):Let me now explain why I think that authors' main claim (namely: that MNLE is more efficient than LAN) may not be strongly supported by the results reported in the current version of the manuscript.The main piece of evidence is summarized in Section 2.2, where authors claim that MNLE learns likelihood function as accurately as LAN, but with only a fraction of the simulation budget. In brief, authors showed that a MNLE estimator trained with 10^5 model simulations is as accurate as a pre-trained version of the LAN method for DDMs (which used 10^11 model simulations). The problem here, is that they did not show that a LAN estimator trained with only 10^5 model simulations yields less accurate results. Or that MNLE trained with 10^11 simulations yields much better results. More generally, what is needed here is a quantitative comparison of the efficiency of the method. One would expect that, for both MNLE and LAN, increasing the number of model simulations increases the accuracy of likelihood approximation. However, this probably follows the law of diminishing marginal utility (i.e. using 10^11 simulations may bring little advantage when compared to e.g. 10^10) (:) Hence, efficiency here should be measured in terms of the speed at which the results accuracy increases. In other terms, claiming that MNLE is more efficient than LAN is important, but requires more work than what is done here. Authors should systematically vary the number of model simulations (spanning, e.g. the 10^5 to 10^11 range), and quantify each method's efficiency in terms of the accuracy profile over this range.Now, this analysis should be performed with different "accuracy" metrics. In particular, authors may measure likelihood approximation accuracy, parameter estimation accuracy and estimation uncertainty accuracy. The latter are critical, because if errors in likelihood approximations may have very small impact on parameter estimation. I note that, w.r.t. parameter estimation accuracy and estimation uncertainty accuracy, I don't think authors chose very informative metrics. For example, the former is defined as the "absolute difference in posterior sample mean normalized by reference posterior standard deviation". If I understood correctly, this metric may show significant differences between MNLE and LAN that would have no practical consequence whatsoever, given that methods would be very similar in terms of their absolute parameter estimation errors. In fact, this is what Figure 4A seems to indicate. I would rather suggest to use accuracy metrics that practically "mean something" for parameter recovery. Measuring parameter estimation error is a possibility (although one may then conclude that MNLE brings no clear advantage when compared to LAN, cf. Figure 4A). But there are other possibilities. For example, likelihood approximation errors may induce pairwise parameter confusions. In turn, for a small simulation budget, LAN estimators may show more pairwise parameter confusion than MNLE. Note: to quantify pairwise parameter confusion in the context of DDM, authors may take inspiration from Feltgen 2021 (or any other metric of their choice).

Thank you for the detailed review and explanation of your concerns with the paper. As mentioned above in the point-by-point response to the essential revision, we were now able to train LANs ourselves and run experiments with smaller simulation budgets. We trained LANs with budgets of 10^{4, 5, 6, 7, and 8}^ and observed, as expected, an overall decrease in likelihood approximation accuracy with decreasing budget. As addressed in detail above and now reported in the updated version of Figure 2, we found that the likelihood approximation accuracy of LAN matched that of MNLE (trained with 10^5^) simulations only for a budget of 10^11^ simulations. The performance with the smaller budgets decreased monotonically with decreasing budgets and resulted in significantly worse performance in likelihood approximations as well as in posterior accuracy, posterior calibration and parameter point estimation accuracy. We clarify the algorithmic differences between LAN and MNLE and the reason for the efficiency difference in the main text (Methods and Materials—Relation to LAN).

We also thank you for the suggestions regarding performance metrics. We updated the posterior accuracy metrics to be in absolute terms and added the parameter estimation accuracy metric as described above. As our additional results confirmed our expectations with respect to the LAN performance on smaller budgets, we did not add a metric on parameter confusion.

Overall, we hope that our additional experiments alleviate your concerns regarding the efficiency of MNLE compared to LAN.